# Iraq Is Moving Forward to Achieve Global Targets in Nutrition

**DOI:** 10.3390/children9020215

**Published:** 2022-02-06

**Authors:** Hind Khalid Sabeeh, Saadulddin Hussein Ali, Ayoub Al-Jawaldeh

**Affiliations:** 1World Health Organization (WHO), Baghdad 10025, Iraq; 2Iraqi Ministry of Health, Nutrition Research Institute, Baghdad 10025, Iraq; dr.saadulddin@live.com; 3Regional Office for the Eastern Mediterranean Region (EMRO), World Health Organization (WHO), Cairo 7608, Egypt; aljawaldeha@who.int

**Keywords:** food insecurity, micronutrient, indicators, Global Nutrition Targets, exclusive breastfeeding, internally displaced

## Abstract

From the 1990s and after 2003, Iraq suffered many difficulties which affected its population negatively in different ways; from embargo to political instabilities, conflicts, and wars, collectively leading to food insecurity especially among the internally displaced people. The Ministry of Health and International Organizations worked collectively to improve the nutritional situation among the most vulnerable groups: children under five, and women in reproductive age. This study aims to review the nutrition situation for Iraq in relation to the Global Nutrition Targets and for Sustainable Development Goals. The data used for comparison was obtained from nationally representative surveys conducted in Iraq from 1996 to 2018, including urban and rural areas for all 18 governorates. Results of these surveys showed a gradual decline in undernutrition indicators for children under five, and an emergence of overweight and obesity, indicating an urgent need for collective action from all sectors and related Ministries regarding malnutrition in its different forms. It also revealed a decline in the prevalence of anemia among the targeted women, but an increase of low birth weight in newborn infants. Exclusive breastfeeding though, is still staggering and in need of urgent action. Iraq is transferring from a generalized state of insecurity to a more secure one, emphasizing the need to strengthen systems for efficient monitoring and evaluation. There is also a need for more recent surveys representing Iraq, as the available data is scant, particularly dietary intake studies within the normal population and internally displaced families in camps or scattered shelters.

## 1. Introduction

The area of Iraq is 435,052 square kilometers [1] and it is one of the easternmost countries in the Arab world [2].

One of the major global public health problems is malnutrition, a direct determinant to the burden of disease. It relates to nutrient intake and can be in the form of deficient intake or an excess. It can present in the form of both undernutrition and overweight/obesity, as well as diet-related non-communicable diseases. The Eastern Mediterranean region (EMR) is no exception [3]. In 2020, around 39 million children less than five years of age were overweight or obese [4]. The weight of disease due to inadequate nutrition continues to grow as stipulated by the Regional Strategy on Nutrition (2010–2019) [5]. The main underlying causes of malnutrition have been shown to pesrsist specifically among infants, young children, adolescents, and women [5].

The period between the 1970s and 1980s in Iraq saw significant and quick improvement in the health indicators, only to deteriorate again after the first Gulf War of 1991 [6]. Iraq was one of the middle-income countries before 1990, and malnutrition was scarce, as a nutritious diet was affordable [7]. Well-equipped, well-supplied, and well-staffed health facilities guaranteed health care services [7].

Nutritional problems began appearing in the early nineties due to factors that led to many health and nutritional problems such as stunting and wasting [7].

In an attempt to counteract this, the Iraqi government relied on distribution of the food basket ration which assured that every citizen received a monthly ration of food items necessary to maintain nutritional requirements [7].

In addition, ongoing humanitarian response in Iraq has been more complicated than ever due to the recent Mosul, Al-Anbar, Salahaldeen, Kirkuk, and Diyala military operations, leading to a huge number of human displacements ([7], p. 9). Government spending on health reduced immensely, leading to destruction of the health infrastructure which resulted in immigration of many of the medical professions [6].

Health care data from the World Health Organization (WHO) confirm that the Iraqi population is paying a crippling price for the continuing violence and corruption within the health system [8].

As a result, thousands of Iraqis remained without access to essential health care due to ongoing security constraints, especially after the 2003 war on Iraq [9].

Nutrition services at the primary health care center levels have deteriorated due to the outbreak of COVID-19 and preoccupation of the majority of health staff with the outbreak.

Iraq is currently on course to prevent the increase of overweight in children under five, but the change in nutritional habits may witness an increased rate of Non-communicable diseases (NCDs) in the future [10].

To reduce malnutrition globally, the World Health Assembly in 2012 endorsed a comprehensive implementation plan for maternal, infant, and young child nutrition, which specified a set of six Global Nutrition Targets that by 2025 aim to

Reduce the number of children under five who are stunted by 40%.Reduce the rate of anemia in women of reproductive age by 50%.Reduce low birth weight by 30%.Prevent childhood overweight from increasing.Raise the rate of exclusive breastfeeding to at least 50%.Reduce wasting in children under five to less than 5% [11].

This article reviews the available proof from data and trends related to nutrition indicators in children under five, in addition to micronutrient deficiencies (anemia) in women, as well.

It can be used as a tool for identifying the data gaps aiming to update the policies and required actions necessary for improving the nutrition situation in Iraq and highlighting the required interventions, to achieve the recommendations of the International Conference on Nutrition (ICN)-2, global targets for nutrition, NCDs, and Sustainable Development Goals [12].

## 2. Materials and Methods

### 2.1. The Search for the Necessary Information Was between 15 August and 10 October 2021

Only nationally representative studies material and English published material were used. Databases from United Nations International Children’s Emergency Fund (UNICEF) on Infant and Young Child Feeding (IYCF) [13], and malnutrition [14] were also used, as were World Health Organization (WHO) resources and governmental websites in Iraq.

National data related to the prevalence of different nutrition indicators was compared to the 2025 WHA targets [15].

Breastfeeding indicators included exclusive breastfeeding, which is defined as feeding infants under 6 months (0–5 months) only breast milk and no other fluids or food [16].

LBW is defined as infants weighing <2500 g at birth regardless of gestational age [12].

Stunting and wasting, were defined as length/height-for-age Z score <−2 standard deviations (SD), weight-for-length/height Z score <−2 SD, respectively, in CUF [17].

Overweight/obesity is defined as any child 6–59 months with >2 Z-scores weight for length/height [17].

WHO–UNICEF criteria for defining thresholds related to stunting, wasting, and overweight/obesity, were used [18].

Nutritional edema was assessed by the presence of bilateral pitting edema in the lower limbs ± body [18].

As for anemia, a hemoglobin concentration of <12.0 g per deciliter for nonpregnant women of childbearing age was considered moderate, and a hemoglobin concentration < 7.0 g/dL was considered severe ([1], p. 44).

### 2.2. The National Surveys from Which Data in This Report Were Obtained

#### 2.2.1. The Multi Indicator Cluster Surveys (MICS) from 1996 to 2018

Sampling for the MICS surveys was stratified cluster sampling ([19], p. 3). These surveys provide data for many indicators regarding children and women at the governorate levels [19].

The MICS survey 2018 utilized Computer Assisted Personal Interviewing Version 6.3. Questionnaires and anthropometric measurements were used for assessment of the nutritional status of targeted children, in addition to clinical examination [17].

#### 2.2.2. Iraq National Micronutrient Deficiencies Assessment and Response Survey 2011–2012 (MNAR)

This was the first national survey carried out for analysis of micronutrient deficiencies including anemia in women 15–49 years, as well as many other nutritional indicators. The survey was a cross-sectional survey of Iraqi households and primary schools [1].

Data were obtained from questionnaires, anthropometric measurements, and collection of biological specimens, including capillary blood collection, from children aged six to 11 months and school aged children six to 12 years for estimation of hemoglobin level. Venous blood was collected from children aged one to five years and non-pregnant women 15–49 years, as well as urine collection for these women, and also for school aged children six to 12 years [1].

#### 2.2.3. Iraq Family Health Survey (IFHS) Report 2006/7

This survey used sampling units similar to the MICS-III 2006 survey. This survey (IFHS) was used as a resource for comparison of data regarding anemia in women aged 15–49 years [20].

Global Nutrition Targets: A tracking tool was developed by WHO to assist countries in monitoring progress towards these goals [15].

Nutritional indicators mentioned above for Iraq were compared to these targets, to track the progress towards meeting the goals set for 2025 [15].

## 3. Results

The results in this manuscript will be presented according to nutritional indicators related to the Global Nutrition Targets and in categories as follows:

### 3.1. Nutritional Growth Assessment Indicators

The Multi-Indicator Cluster Survey (MICS1; 1996) revealed the prevalence of wasting equivalent to 11%, while stunting was at an average of 32% [21].

The following MICS surveys (MICS3 and 4; 2006 and 2011, respectively) [19,22] revealed a drop in these figures; 4.8% and 21.4% for wasting and stunting, respectively (MICS3) [22].

According to the MICS4 2011 indicators [19], every one in four children experienced stunted growth (22.6%), and 7.4% of children under five experienced moderate and severe malnutrition (wasting) (Figure 1 below).

The latest Comprehensive Food Security and Vulnerability Analysis 2016 showed that stunting occurred at low levels of severity with rates at 16.6% in resident children and 19.2% in children of internally displaced people (IDP). Wasting was in the medium se-verity range, with 7.5% for residents and 5.5% for IDPs. Previous comprehensive food security and vulnerability analysis surveys (CFSVAs) revealed higher malnutrition rates in males compared to females, although the differences were not dramatic [23].

Based on the latest MICS6 2018 indicators, the rate of stunting declined to 9.9%, and 2.5% for wasting. Figure 1 and Figure 2 below show that the trend of malnutrition rates in children under five years gradually decreased throughout the years, reaching acceptable levels [17].

Looking at this figure, wasting is found to be highest at zero to five months, and stunting between the ages of 24–30 months. (Figure 3 below) [17].

Figure 4 below shows a strong correlation of the mother’s educational level with malnutrition, but not so for the household wealth index [17].

In general, Table 1 (below) shows malnutrition rate trends obtained from MICS surveys which show fluctuating rates regarding malnutrition indicators that are harmonized with demographic and political changes in Iraq and interventions initiated accordingly ([7], p. 10). The indicators from 1996 were also included in this review as we follow the WHO global analysis to show progress since 1990 (WHO, WB, and UNICEF data base).

Regarding childhood overweight in CUF (Figure 5 below), MICS6 2018 reveals that the national average is 6.6%, higher in males and in the urban residents, while Figure 6 reveals that overweight is highest around the age of 12 months [17].

### 3.2. Low Birth Weight (LBW) in Newborn Infants

In Iraq and according to both MICS surveys (2011 and 2018), Table 2 reveals that the rates of low birth weight increased from 13.4–25.2%.

### 3.3. Exclusive Breastfeeding

In Iraq, Table 3 below reveals the difference in exclusive breastfeeding rates found in the two Multi-Indicator Cluster Surveys (2011 (18.6%) and 2018 (26%)). There is an obvious increase, but not to the desirable level of at least 50%.

The survey carried out in March 2021 in Baghdad City for mothers attending primary health care centers revealed that 53.5% of mothers were practicing exclusive breastfeeding at the time of the survey [24].

### 3.4. Micronutrient Indicators: Anemia in Women of Reproductive Age

The Non-communicable Disease Department at the Ministry of Health in Iraq carried out a survey in 2006/7, which showed anemia prevalence among non-pregnant, non-lactating women in the abovementioned age group equivalent to 35.5%, and it even reached 40% among women between 40–49 years of age ([20], p. 23).

Six years later, in 2013, the Nutrition Research Institute carried out the MNAR Survey, which revealed a drop in this indicator to 19.9%, and that indicators for deficiency in iron, and anemia due to iron deficiency, were 24.5%, and 4.9%, respectively, as seen in Table 4 below [1].

Older women, compared to younger, were more likely to suffer from this ailment as were married, compared to unmarried, and were more iron deficient, with iron deficiency anemia as well. Anemia was also found to be higher among women from southern regions compared to other areas in Iraq. Urban-residing women were more anemic than their rural counterparts and with a higher percentage of iron deficiency anemia as well [1].

This survey revealed an almost equal distribution of deficiency in vitamin B12 levels among age groups and was equal to 29.3%, but higher in the Kurdistan region, and in the rural areas [1].

Folate deficiency in Women of reproductive age (WRA) was 19.0%. A higher percentage was found in women 15–19 years. Baghdad region had the highest prevalence of folate deficiency (28.6%) compared to other regions in Iraq [1].

Iraq falls under the mild category group for this disease, conforming to the WHO classification [1].

## 4. Discussion

### 4.1. Nutritional Growth Assessment Indicators

#### 4.1.1. Wasting Requires Urgent Attention Owing to Its Associated Risks for Morbidity

Being both stunted and wasted confers an even higher risk of mortality [25] and in Iraq, reviewing different Multi-Indicator Cluster Surveys, we find that there is a gradual decline in the prevalence of wasting, especially so between 2011 and 2018.

Regarding this indicator and relying on the WHO anthropometric cutoffs [26], Iraq dropped from high category to low in the period from 1996 to 2018 and has met the WHA target for 2025 regarding wasting (<5%) [10,11].

Comparing this rate of wasting in Iraq, it is similar to that in Kuwait (2.5%) in 2017, and much lower than the rate in both Oman (9.3% (2017)) and Syria (11.5% (2010)) [27].

This was due to the active procedures taken by the Nutrition Research Institute (Ministry of Health/Primary Health Care Department) through capacity-building of the nutrition focal points in Baghdad and the peripheral governorates by implementation of intensive training courses and workshops, in addition to printing guidelines for growth monitoring for children under five, counseling, and educational leaflets, which were distributed to the different primary health care centers, with the support of international agencies working in the country. There was also a shift from prolonged embargo and emergency situation to an almost stable life, with an improvement of the income and living conditions of the people due to re-export of oil, which had a positive impact on the food consumption pattern in general.

#### 4.1.2. Poor Nutrition in Utero and Early Childhood Results in Stunting

Upon reviewing the MICS surveys from 1996 to 2018, and using the same WHO thresholds, Iraq appears “on course” to meeting the target for stunting [10], as this indicator dropped from the very high category in 1996 to the medium category in 2018.

Stunting in Iraq shows a lower value compared to the global average of 22% [12], also lower than the average for the Asia region (21.8%) [10], but higher than Jordan (7.7%) [12].

The drop in stunting rates from 1996–2011 was gradual as stunting is a chronic condition and requires new generations with good balanced nutrition to see a further decline [8]. A sharper decline was seen between 2011 and 2018. Improvement of the socioeconomic conditions as well as improved nutrition were attributable factors.

#### 4.1.3. Overweight Is a Main Contributor for Non-Communicable Disease

According to the prevalence thresholds of public health significance regarding overweight (WHO) and comparing results of overweight in CUF in Iraq between MICS4 2011 (p. 23) and MICS6 2018, the national standard declined from a high threshold to the medium category for this indicator [17,19,26].

This rate though, was higher than the global average of 5.7% and the average of Jordan (4.7%) but was close to the average rates of EMR (7.7%) [12].

If this rate continues to climb, Iraq will fail to achieve the WHA target for 2025 [11].

Although the Iraqi Ministry of Health is working through its different departments to promote a healthy diet and lifestyle to combat obesity and overweight, there is an obvious change in the diet and lifestyle favoring consumption of energy dense foods, beverages sweetened with sugar, processed foods, and adherence to more sedentary behaviors [12], which may predict an increase in overweight/obesity among all age groups in Iraq and not just those under five in the MICS surveys to be conducted in the following years.

### 4.2. Anemia in Women

Anemia has negative impacts on national economies, and in pregnant women is associated with higher mortality and morbidity outcomes in both the mother and infant ([28], p. 25).

This review revealed that the rate of anemia among WRA in Iraq declined from the moderate to mild category (WHO standards) within six years and is nearing the WHA target of 15% [29].

Potential factors responsible for anemia among WRA could be different nutritional deficiencies [12]. Although almost one fourth of WRA were found to have iron deficiency (MNAR), only 5% had iron deficiency anemia, while almost one third (30%) had vitamin B12 deficiency and almost one fifth (19%) had folate deficiency [1].

A wheat flour fortification program was introduced in Iraq in August 2006 and has been intermittently struggling to fulfill the need to fortify wheat flour with iron and folate since then, in order to meet the World Health Organization (WHO) recommendations for fortification ([7], p. 9). The aim of this intervention was to reduce the burden of anemia.

In addition to the fortification program, and in the attempt to reduce anemia among WRA, the Ministry of Health set up a program promoting pregnant women to utilize the antenatal care services at the primary health care level routinely throughout pregnancy where they are provided with folate supplements in the first trimester and supplements containing iron and folic acid from the beginning of the second trimester throughout the remainder of pregnancy and until three months after delivery ([1], p. 14).

This review reveals that the persistence of anemia is still a challenge. In spite of this, and although there was less than 50% drop in the prevalence of anemia in these women, it definitely denotes an overall improvement in women’s health [1], most likely due to a healthier diet from improved socioeconomic conditions and improved salaries.

### 4.3. Low Birth Weight (LBW)

Neonatal Deaths Are Found to Be Much Higher in LBW Newborns [23]. The necessary data to assess whether Iraq will achieve the 2025 goals Is lacking at the moment [10]. The data in this study showed a rise in the LBW rate, which imposes a challenge in Iraq and has exceeded the 2015 global average (14.6%) [12].

Although the percentage of low birth weight in Iraq increased, comparing the MICS 4, 2011 and MICS 6, 2018 results, the data (Table 2 above) shows that in 2011, only about half (52.6%) of newborns had been weighed after birth, while in 2018, this percentage increased to almost three fourths of newborns (72.0%).

The number of institution deliveries increased, comparing data from the MICS4, 2011 survey and the MICS6, 2018 survey (74.4% and 86.6%, respectively), as did the percentage of deliveries attended by a skilled birth attendant (88.5% and 95.6%, respectively) [17,19].

The Ministry of Health had implemented intensive training courses and workshops for health care workers in the field of newborn care, and recently developed new strict protocols and procedures to be followed in the delivery room which were monitored regularly, one of which was weight of the newborn, and due to the higher number of institutional deliveries attended by a skilled birth attendant in 2018, this could explain the difference in the findings and that the actual number of LBW infants in 2011 may have been underestimated.

### 4.4. Exclusive Breastfeeding

The World Health Organization recommends that infants be exclusively breastfed until they complete 180 days of age, benefitting the health of both infant and mother [30].

Increasing rates of exclusive breastfeeding will assist achievement of World Health Assembly (WHA) Global Nutrition Targets and is critical to include in the Sustainable Development Goals framework [12].

There has been progress towards achieving the exclusive breastfeeding target in Iraq [10], but it is evident that the overall status of this indicator among infants in Iraq is not to the required levels.

Upon comparing results of rates of exclusive breastfeeding obtained from the MICS surveys 2011 to 2018 (Table 3 above), there is an increase in the prevalence of EBF, but it is lower than the global average (44%), and the estimate reported for the EMR (Jordan, 34%), while also being below the WHA target of 50% by 2025 [12].

The elevation in the rate of EBF during this seven-year interval was achieved by actions taken by the Nutrition Research Institute through supporting IYCF services in programs that aim at increasing the number of baby-friendly hospitals from the north to the south of Iraq, by placing a yearly plan for this activity, in addition to capacity-building through implementing intensive workshops and training courses on IYCF counseling for all nutrition health workers in Primary Health Care Centers (PHCCs) from the north to the south of Iraq. International agencies funded travel abroad for responsible personnel in this field as well.

The Ministry of Health formulated information material which, with the support of UNICEF, was printed out and distributed to all PHCCs to hand out to WRA attending these centers to raise awareness regarding benefits of breastfeeding and disadvantages of breast milk substitutes (bottle feeding) in addition to timely complementary feeding.

The joint action of both the Ministry of Health and Justice to issue local national regulations related to breastfeeding and breastmilk substitutes based on the Code of Marketing in the form of “Instructions No. 2 for Promotion and Protection of Breastfeeding” in 2015 was a major step to support breastfeeding. The main drawback was the absence of a penalization system for violators.

Advocacy of breastfeeding through producing video clips on breastfeeding with support from UNICEF was carried out by participation of famous socially accepted Iraqi actors (champions), and it also contributed to raising awareness.

## 5. Conclusions and Recommendations

More up-to-date studies regarding food consumption habits are required to prevent the upward climb of the rates of overweight/obesity within the Iraqi society, starting from a young age.

The Ministry of Health should aim at interventions to increase awareness among the population regarding optimal dietary choices for families and a change of the current lifestyle behaviors. Policies should be set targeting food systems to produce healthier foods, including food reformulation laws and labeling regulations [12] in addition to a monitoring system to regulate importing unhealthy food.

The status of LBW is of concern given its potential consequences on impairing growth, cognitive development, and immunity among other adverse health effects [12]. The obvious increase in the rate of LBW in Iraq reflects the need for governmental interventions and a strong monitoring system. To help improve LBW rates, maternal health and nutrition requires more attention in addition to investigating the underlying causes.

As for anemia, regarding iron, folate, and B12 deficiency, as well as other micronutrient deficiencies, future research studies targeting the underlying causes of micronutrient deficiencies are essential.

However, to establish a more accurate interpretation of the role of the fortification program on iron status, additional data on the actual dietary intake is needed [12].

Recent analyses have documented that increasing rates of breastfeeding could help to foster smarter, more productive workers and leaders [31].

In order to improve the low rates of exclusive breastfeeding in Iraq, it is the responsibility of the Health Ministry to strengthen national related regulatory systems, especially towards companies importing breast milk substitutes. There should be a penalizing system for violators of the Iraqi Instructions and Code.

Building and reinforcing the skills of health professionals to implement the required policies effectively to promote and protect breastfeeding is a priority.

It is vital to also integrate educational materials into the academic medical curricula to increase awareness of the public health aspect of breastfeeding.

Providing community-based services, to help breastfeeding mothers discharged from delivery facilities face the challenges they may meet after returning home and throughout their second year postpartum, is an essential requirement and not yet available in Iraq [19].

A proper assessment of the nutritional status among vulnerable populations and in camps for internally displaced people should be implemented regularly, in addition to evaluating food availability and security, especially for children and women.

There is also a very strong need for the government to take positive action politically and permit all internally displaced families to return to their residence of origin and lead natural lives, as the majority are farmers who lived on the land.

As Iraq is painfully slowly transferring from a generalized state of insecurity to a more secure one, with reduction of internal fighting, there is also a need to further strengthen systems operating in nutrition and conducting updated surveys in this domain to gather more informative data.

This was a brief description of the nutrition situation analysis for Iraq showing progress towards the 2025 global targets and sustainable developmental goals (SDGs).

## Figures and Tables

**Figure 1 children-09-00215-f001:**
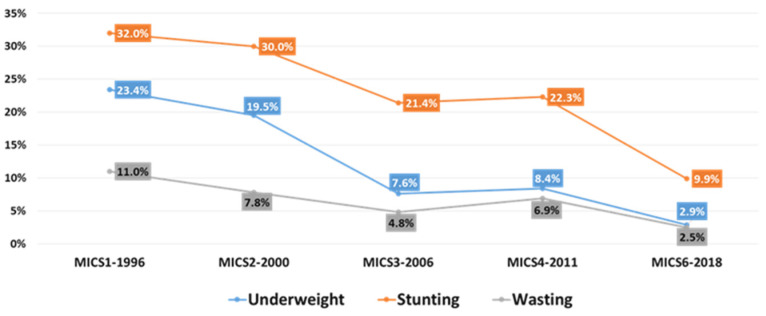
Malnutrition rates in children under five in Iraq (MICS6-2018).

**Figure 2 children-09-00215-f002:**
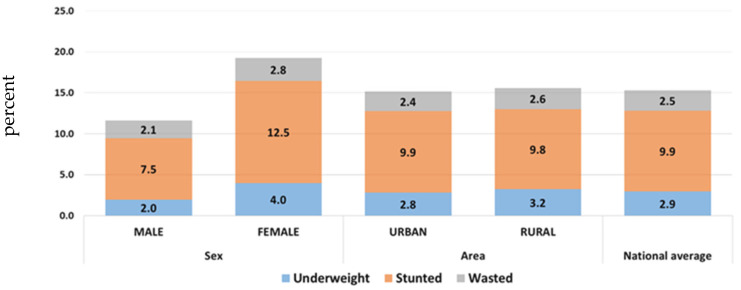
Malnutrition rates in children under 5 years by sex and geographical location (Iraq, MICS 6-2018).

**Figure 3 children-09-00215-f003:**
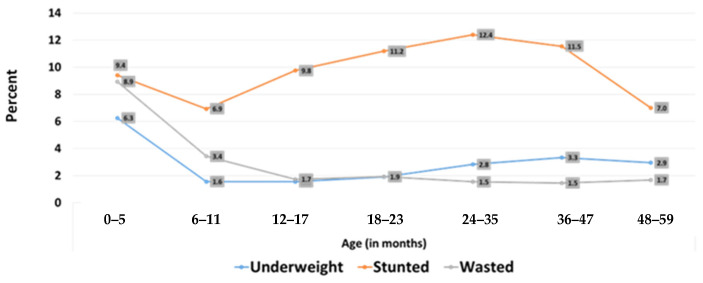
Malnutrition rates in children under five years of age in months (Iraq, MICS 6-2018).

**Figure 4 children-09-00215-f004:**
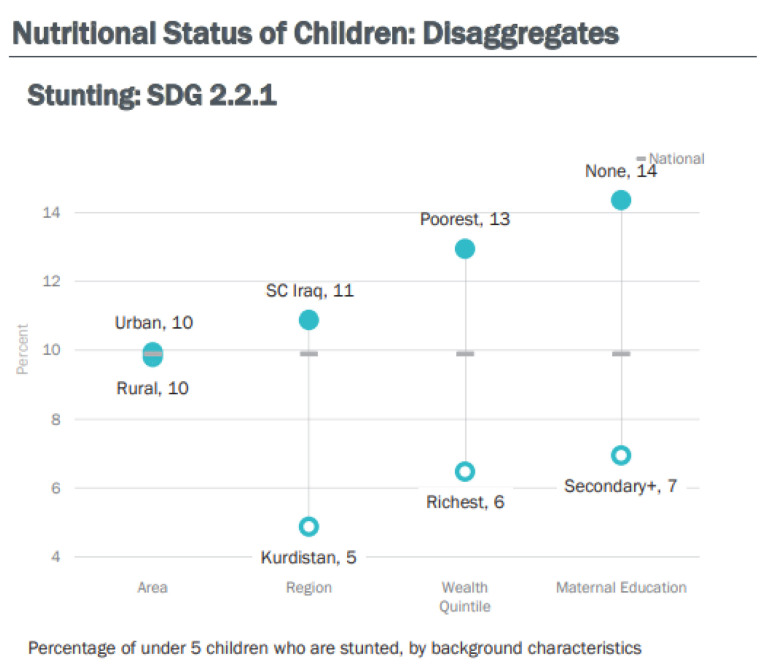
Percent of children under five who are stunted, according to background characteristics (Iraq, MICS 6-2018).

**Figure 5 children-09-00215-f005:**
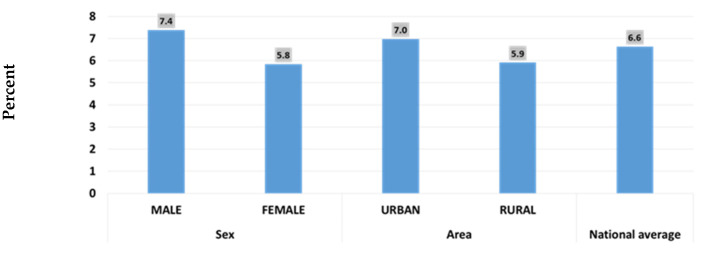
Overweight in children under five years (Iraq, MICS 6, 2018).

**Figure 6 children-09-00215-f006:**
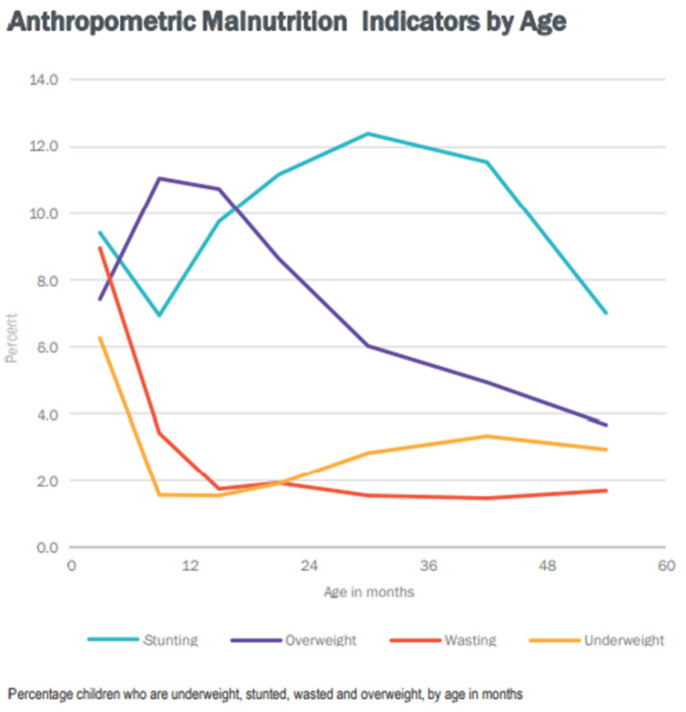
Anthropometric malnutrition indicators by age source: Iraq, MICS 6, p. 25.

**Table 1 children-09-00215-t001:** Malnutrition rates by years based on different MICS surveys for Iraq in children under five.

Survey	Malnutrition Rates		
	Underweight	Stunting	Wasting
MICS1-1996	23%	32%	11%
MICS2-2000	19.5%	30.0%	7.8%
MICS3-2006	7.6%	21.4%	4.8%
MICS4-2011	8.4%	22.3%	6.9%
MICS6-2018	2.9%	9.9%	2.5%

**Table 2 children-09-00215-t002:** Percentage of low birth weight according to MICS Surveys (Iraq, 2011 and 2018).

MICS Survey	Total Number of Newborns Weighed after Birth	Percentage of Low Birth Weight
MICS 4 2011	52.6%	13.4%
MICS 6 2018	72.0%	25.2%

**Table 3 children-09-00215-t003:** Percentage of exclusive breastfeeding in Iraq (MICS surveys, Iraq).

MICS Survey	Percentage of Exclusive Breastfeeding
MICS 4 2011	18.6%
MICS 6 2018	26%

**Table 4 children-09-00215-t004:** Anemia indicators in nonpregnant women. MNAR Survey 2012, Iraq.

Topic	Indicator
Anemia	19.9%
Iron deficiency	24.5%
Iron deficiency anemia	4.9%

## Data Availability

Not applicable.

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
