# Peer review of "Iraq Is Moving Forward to Achieve Global Targets in Nutrition"

_children, 2022, doi:10.3390/children9020215_

Round 1
Reviewer 1 Report
why the authors included 1996 data? - It seems to me that Iraq meet the MDG (from 200-2015) and now on track for SDGs
The aim is different from the title rather, the aim should reflect the title. If the aim was "This study aims to describe the trends of malnutrition for children under five and micronutrient deficiencies (anemia) among women 15 to 49 years"
one would like to see a trend analysis?
undernutrition has reduce significantly between 2011 and 2018 - why?, any intervention done in Iraq between these periods?
The figures title should include Iraq
I don't understand Table 6 and it could be included as text.
there are no statistical sections indicating how these values were got - were the values taken from the report?
how did the authors categorised the outcome variables used in this study?
the authors need to explain why there are gradual decline in undernutrition indicators for children under five? if low birth weight has increased during the same period
Conclusion:
I would the authors to start their conclusion by summaring their key findings based on the research aims
What are the policy implications of the authors findings especially increased low birth weight and low percentage of breastfeeding (suboptimal)? - also suggest intervention to reduce or increase these variables.
Author Response
Good evening.
1- Reply to comment 1:
We included data from 1996 because we are following the WHO global analysis to show progress since 1990 (WHO, WB and UNICEF data base).
2- Reply to comment 2: The aim was changed in the new manuscript to:
"This study aims to review the nutrition situation for Iraq showing its progress towards achieving the Global Nutrition Targets and Sustainable Development Goals".
3- Reply to comment 3:
The analysis was put in the discussion.
4- Reply to comment 4: Why undernutrition decreased between 2011 and 2018:
Many factors were involved;
a- Improvement in the socioeconomic condition and salaries due to export of oil once again. Employees before that period could not buy any nutritious food items like meat or eggs or any dairy products. Their diets were mainly staple substances (rice, bread).
b- The Ministry of Health took steps to implement intensive training courses and workshops on anthropometric measurements, advocacy of healthy diet, management of SAM in CUF, IYCF counseling, and other nutrition subjects in country and abroad.
c- There was a shift from emergency state and embargo to an almost stable life, so people could work and make money.
d- Gradual improvement of the security situation.
5- Reply to comment 5: The figures title should include Iraq:
It was done, and all figures title included Iraq.
6- Reply to comment 6: Table 6
Table 6 was how WHO/UNICEF would categorize the public health problem of the included indicators (wasting, overweight and stunting) in CUF and this specific table was taken out as a table and included within the text.
7- Reply to comment 7: Values in the report:
All values in the report were findings and results of nationally representative surveys.
8- Reply to comment 8: How were outcome variables categorized:
The outcome variable were categorized by using the new WHO/UNICEF Prevalence thresholds which had a public health significance for the involved indicators.
9- Reply to comment 9:
There was gradual decrease (improvement) in undernutrition indicators for the same reasons mentioned in reply number 4, but the values for LBW seemed to increase, as the MOH put strict guidelines for health workers providing services in the delivery room which were strictly monitored and evaluated at fixed time intervals and so all new borns were weighed immediately after birth and their weight was written in the delivery room forms. This was not so closely monitored beforehand and sometimes taking weight of the newborn was neglected, forgotten or estimated roughly. This was entered in the changed manuscript. For proper follow up of this indicator further surveys should be applied in this field.
10- Conclusion:
a- The key findings in the Conclusion section were summarized based on the aims of the study.
b- Both variables; low birth weight and low exclusive breastfeeding rates need to be improved in the years to come. There should be a penalizing system in place for violators of the Code and an increase in BFHs. This requires strong support from the MOH and other stakeholders.
There should be an increase in comprehensive training courses and workshops in both fields of IYCF and Care of the newborn and Maternal care and stress on good maternal nutrition and health pre-pregnancy and during pregnancy.
All hospitals should abide to the BFHI regulations even if they do not at the moment attain the title and alliance to the BFHI regulations should be a must for any new hospital agreements.
Reviewer 2 Report
The paper by Hind Khalid et al. on the Nutrition Situation Analysis for Iraq Showing Progress To-2 wards Global Nutrition Targets and Sustainable Development Goals contains interesting data on the nutrition of children and women in Iraq
Major points:
- Line 117: How was ethical approval achieved. To give a reference is not enough
- In Table 1: which group of children is described? Children under 5 or the whole cohort?
- Line 298: How is LBW defined? Should be in the method section.
- 1.2: In this section results are given again, which reflects a repetition. The results need to be discussed
- 1.3: What about the results from other countries in your area?
- 1.4: Again, repetition of results. They need to be discussed with other results
- Table 6 and 7 should be displayed in the results, not in the discussion
- 2: Again, results are given rather than being discussed
Minor points:
- Typo in line 125
- Typo in line 282
Conclusion: The discussion of the paper should be rewritten and shortened in the way that repetitions of results should be avoided. Instead the results need to be discussed with other published data.
Author Response
Good evening. Please find below reply to your comments:
1- Method and Result Sections:
Reply: The methods and the result sections have been improved.
2- Conclusions:
Reply: The conclusion section was improved and supported by the results in the new manuscript.
3- Ethical Approval:
Reply: Ethical Approval was achieved by appointing a committee of officials from the different ministries involved in the survey. These officials would be chosen by the responsible Director Generals. This committee would be responsible to review the aim of the survey, the questionnaires, the methodology and tools used, the acceptability of the population to participate in answering the questionnaires, and time of implementation of the survey. They would change what they felt was inappropriate and then give the green light to start. There would be an official document for the names of the officials in this committee signed by the Director General with a number and date.
4- Table 1:
Reply: Table 1 involves children under five.
5- definition of LBW:
Reply: LBW is defined as any newborn weighing less than 2500 grams regardless of gestational age and, yes it is now in the Methods section.
6- Repetition of results:
Reply: Sorry about repetition of results. I agree with you. Results are now confined in the "Results" section and not repeated in the other sections, but discussed in the "Discussion" section (without repetition).
7- Results from other countries:
reply: This has been done by providing data for other countries and results in this study were compared to that data.
8- Tables 6 and 7:
Reply: Table 6 was taken out as a table and used within the text as reference for concluding the public health situation of Iraq regarding the related indicators. As for table 7, only the national values were used.
9- Discussion of results:
Reply: They have been discussed and not repeated in the discussion section.
10- Typos:
Reply: Corrected.
11- Conclusion:
Reply: the manuscript has been rewritten in a better way and shortened as you have mentioned here, AND repetitions of results has been cleared and taken care of. Other published data have been included and compared with the data from this study.
Thank you very much for your review, and I hope I have answered your comments in a proper way.
Best regards
Round 2
Reviewer 2 Report
- The reply of the authors to the ethical approval is ok, but does not appear in the text of the manuscript. It definetely need a description of the ethical approval within the text of the mansucript.
- The other points have been delt with satisfaction
Author Response
Dear Reviewer 2,
Good day.
I pray this finds you well. Thank you for your review.
1- Reply to comment 1: I will rewrite this section within the text in more detail as you have requested.
2- Reply to comment 2: We appreciate your review.
Thank you and best regards,
Dr. Hind